# Photo-initiated solvent-mediated depolymerization of consumer poly(methyl methacrylate) without chlorinated reagents

Jonathan T. Husband [1,2] ✉, Gavin Irvine [1,2], Callum R. Morris[3], Andrea Folli [3,4], Matthew G. Davidson [1,2] ✉ & Simon J. Freakley [1,2] ✉

The chemical recycling of commodity acrylic polymers, such as the transparent thermoplastic polymethyl methacrylate (PMMA), typically requires temperatures of 350-400°C. Herein, we report chemical recycling back to monomers for PMMA between 120-180°C, through UV illumination under oxygen-free conditions. We have achieved gram-scale degradation of consumer plastic with >95% conversion, yielding >70% monomer, which can be readily repolymerized. The process proceeds even at high concentrations (>1 M) and depends strongly on solvent choice: aromatic solvents like dichlorobenzene and diphenyl ether maximize conversion. In contrast to a concurrently published study, we report that chlorine radicals are not required for depolymerization; however, when present, they react with the unzipping chain to form chlorine-functionalized PMMA which can be upcycled through derivatization. In more sustainable non-chlorinated solvents such as benzonitrile, minimal termination by radicals enables complete unzipping. These findings demonstrate a low-temperature, scalable route for the chemical recycling of PMMA, offering alternative pathways for plastic circularity.

Plastic materials are ubiquitous in society due to their tunable mechanical and chemical properties while being processable with lower energy and resource input relative to other materials, such as glass and aluminum[1]. As such, there is a significant amount of carbon "locked" into commodity plastics that could be a sustainable feedstock for chemical processes without the need for the extraction of virgin fossil-derived carbon. Effective end-of-life treatment of commodity-scale plastics is also a pressing environmental concern. The inherent chemical stability of these polymers results in slow biodegradation, microplastic generation, and chemical contamination of the environment, leading to ecosystem damage[2].

Mechanical recycling is widely used to utilize this source of carbon and reduce environmental leakage; however, it can be intolerant to contamination, and it is progressively detrimental to polymers' thermomechanical and visual properties[3]. Chemical recycling is often touted as the solution to these issues[4], as it provides the opportunity to turn lower quality recyclate into a fresh monomer stream that can be purified and then repolymerized for formulation into plastics identical to virgin materials. However, excluding the chemolytic or selective chemical recycling techniques for polyesters and similar plastics[5-8], chemical recycling to monomer of commercial vinyl plastics currently relies upon thermolysis, where high temperatures in the absence of oxygen are used to homolytically break polymer bonds[9]. In some vinyl polymers such as polystyrene and polymethyl methacrylate (PMMA), the radicals formed are relatively stable, which leads to chain unzipping and depolymerization, often referred to as monomer recycling[10,11]. PMMA, also known as Perspex® and Plexiglass®, is predicted to be produced on a ~6 million tons p.a. scale by 2028 with diverse uses across the medical, automotive, and construction industries[12]. Current monomer recycling of PMMA relies upon high

[1]Bath Institute of Sustainability and Climate Change, Bath, UK. [2]Department of Chemistry, University of Bath, Bath, UK. [3]Cardiff Catalysis Institute, School of Chemistry, Cardiff University, Translational Research Hub, Cardiff, UK. [4]Net Zero Innovation Institute, Cardiff University, Translational Research Hub, Cardiff, UK. ✉e-mail: jth67@bath.ac.uk; chsmgd@bath.ac.uk; sf756@bath.ac.uk

temperatures and energy inputs (>350 °C)[13], resulting in a large carbon footprint for circularity. Therefore, new lower-temperature technologies are urgently needed to reduce energy requirements and make chemical recycling more commercially competitive and sustainable.

To date, reports of lower-temperature chemical recycling of such polymers have been almost exclusively limited to polymers synthesized by controlled radical polymerization (CRP) or processes that leave accessible chain functionality, such as alkenes for metathesis depolymerization[14,15]. In the absence of such accessible functionality (e.g., for commodity PMMA materials), high temperatures are needed to initiate polymer chain radical formation. A number of studies have recently explored the optimization of lower-temperature processes on CRP-synthesized polymers[16,17]. Examples of low-temperature depolymerization systems include iron and copper-mediated atom transfer radical polymerization (ATRP)[18–20], thermally driven reversible addition-fragmentation chain transfer depolymerization (RAFT)[21], including in flow[22], and photocatalyzed depolymerization of RAFT polymers (Fig. 1)[23]. A thorough review of such techniques has been published by Anastasaki et al.[16].

Due to its relatively low ceiling temperature ($T_c$), a significant number of these reports are focused on polymethacrylates, in particular PMMA. Some of the most promising progress has been reported by Sumerlin et al., who have developed the bulk depolymerization of PMMA at temperatures ~200 °C by synthesizing RAFT and phthalimide functionalized methacrylates[24]. Furthermore, the group has shown that decarboxylation of PMMA backbones can lead to radical formation and subsequent degradation back to a mixture of products, including monomers in low conversion[25]. In addition to end-chain modified polymers, several groups have facilitated efficient conversion to monomer through the incorporation of co-monomers, which can be used to introduce backbone radicals[26]. For example, the incorporation of a pendant boryl group into the backbone can be used to achieve depolymerization in response to acid cleavage[27] and the low ceiling temperature of α-methylstyrene has been utilized to enable efficient depolymerization in copolymers with MMA[26]. Most recently, Stache et al. have shown the conversion of commercial PMMA and polystyrene without the need for polymer functionality[28,29]. They achieved this by making thin-film composites of PMMA with carbon quantum dots (CQDs), which can be converted to monomer at around 60% conversion through irradiation with UV light. This is attributed to a photothermal effect whereby the CQDs absorb light and transfer thermal energy to the polymer chains, providing the energy for scission and depolymerization.

Inspired by the work of Anastasaki, Stache, and Sumerlin, we hypothesized that UV photocatalysis, such as hydrogen atom transfer (HAT) reagents, could create carbon-centered radicals capable of initiating depolymerization at lower temperatures. Contemporaneously with this study, Anastasaki and coworkers developed PMMA depolymerization in a range of chlorinated solvents, measuring high polymer degradation by size exclusion chromatography (SEC)[30]. Their study focused on chloroaromatic solvents, and postulates a HAT mechanism in which Cl• radicals generated from the solvent abstract hydrogen from the polymer backbone, initiating the unzipping, and concludes: "a chlorinated solvent is indeed necessary for main chain scission to occur." In this study, we observe the same phenomena with chloroaromatics, but expand the process to also utilize a range of non-chlorinated aromatic solvents, with greener potential. We also show mechanistically that the depolymerization behavior is different between these non-chlorinated solvents and those previously reported. This understanding is utilized to undertake functionalization of part depolymerized chains, which we show are recapped by Cl• radicals before full depolymerization can occur, suggesting a route towards partially controlled depolymerization. Combined, these findings show the potential to introduce new functionality on produced oligomers and show advances towards alternative routes of recycling and upcycling consumer vinyl plastics without the need for chlorinated solvents.

## Results and discussion

Initially, our studies investigated the use of an *N*-functionalized amide HAT catalyst reported by Alexanian et al.[31] to initiate depolymerization of PMMA. Dilute solutions of commercial PMMA (25 mM monomer unit concentration, 15 kDa, $M_w^{SEC} = 15.3$ kg mol$^{-1}$) were added to the catalyst at 140 °C (Fig. S5), conditions which favor depolymerization to monomer of RAFT and ATRP functionalized PMMA[21]. Dichlorobenzene (DCB) was selected as a reaction solvent primarily due to its mild polarity, high boiling point, and lack of abstractable protons[32]. The dilute solutions of PMMA in DCB (25 mM monomer unit concentration) were purged with argon, heated to 140 °C, and exposed to UV LED illumination (365 nm, 30 W). Monitoring the reaction at intervals by $^1$H NMR spectroscopy showed the reaction proceeded rapidly with monomer generation before plateauing at 5 h and approximately 20% conversion to monomer. However, in studying the effect of different equivalents (1–10 molar equivalents) of HAT catalyst, it was observed that the reaction proceeded independently of catalyst concentration (Fig. S6). Therefore, a repeat of the conditions in the absence of a catalyst was undertaken, which, surprisingly, also proceeded to 20% conversion to monomer.

Satisfied that UV irradiation without catalysts or additives was sufficient to facilitate depolymerization under these conditions, we

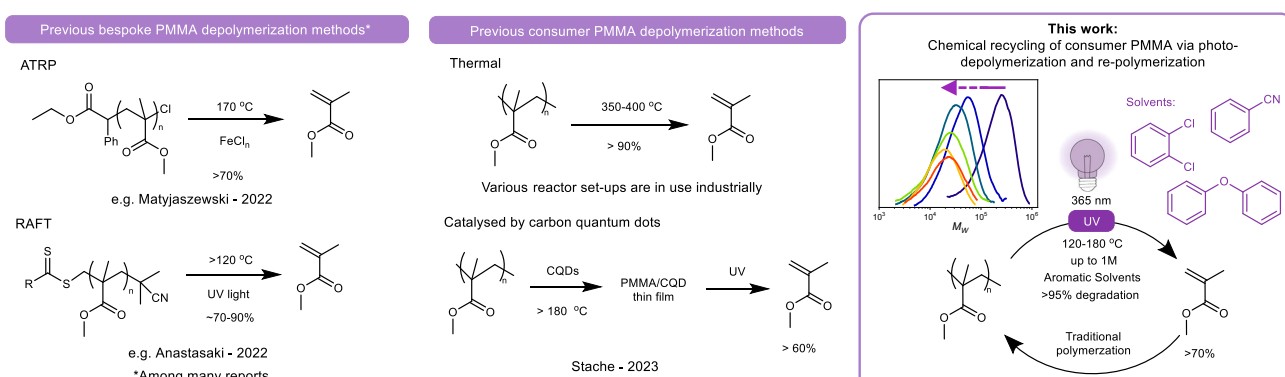

**Fig. 1 | Strategies for the chemical recycling of PMMA.** Left: Example depolymerizations of PMMA with existing handles for depolymerization made by atom transfer radical polymerization (ATRP) and reversible addition-fragmentation chain transfer polymerization (RAFT)[19,21]. Middle: Prior depolymerization methods for commercial PMMA via thermal (top) and photothermal (bottom) processes, the latter employing carbon quantum dots (CQDs)[28]; Right: Our reported system for low-temperature consumer PMMA depolymerization using UV irradiation in aromatic solvents.

investigated the effects of concentration and temperature to optimize conversion. As depolymerization is driven by $T\Delta S$, increasing temperature, increasing the molecular weight of the polymer, and decreasing concentration should lead to improved conversion. Lowering the temperature to 120 °C led to decreased conversion (<10%) and increasing to 175 °C gave conversion to monomer of over 50% at 5 h for the initial 15 kDa polymer (Fig. 2b). As expected, increasing polymer concentration led to a decrease in conversion (from around 40% at 100 mM monomer unit concentration to 20% at 400 mM). To study molecular weight effects, commercial PMMA of ~200 kDa ($M_n^{SEC} = 193$ kg mol$^{-1}$), a molecular weight more representative of PMMA in materials and waste, was tested at the highest conversion concentration (25 mM repeat unit concentration) (Fig. S25). This led to the highest conversion to monomer observed in this study, reaching 72% in 5 h at 175 °C (Fig. 2b). It is important to note, 72% conversion to monomer by [1]H NMR spectroscopy, correlated to >95% degradation to small molecules (<1000 Da) by SEC analysis, the method used by Anastasaki et al. for depolymerization conversion[30]. Promisingly, at this molecular weight, increasing concentration appeared to have a less detrimental effect on conversion, with 400 mM and 1 M both reaching around 60% conversion to monomer.

An identical control reaction, but without UV irradiation, showed no conversion. To gain further insight, a simple on/off irradiation experiment was devised, in which the polymer was subjected to alternating 15-min intervals of UV irradiation and dark. By sampling for [1]H NMR and SEC analysis, it was evident that quantifiable monomer generation and concomitant molecular weight decreases only occurred during periods of UV irradiation, indicating a lack of radical initiation or propagation in the absence of irradiation (Fig. 2d). However, depolymerization could be resumed after periods of dark, showing re-initiation of chains. A concurrent report indicates that 405 nm light-mediated depolymerization of PMMA required chlorinated solvents[30]. To test whether wavelength might be the cause of this discrepancy, 200 kDa PMMA was irradiated with 365 and 405 nm light at 140 °C in DCB and benzonitrile. Interestingly, no significant differences in conversion to monomer were observed, with ~50% for DCB and ~30% in benzonitrile at both wavelengths, showing that in this range, wavelength effects are minimal and non-chlorinated solvents also facilitate depolymerization at 405 nm (Table S1).

SEC monitoring during the uninterrupted depolymerization of 15 kDa PMMA indicated a classical uncontrolled unzipping mechanism with minimal change in $M_w$ or dispersity (Fig. 2e). However, monitoring of the 200 kDa polymer suggested a different pathway, exhibiting a significant and gradual decrease in $M_w$ with minor changes to the polydispersity and RI intensity over the initial period (Fig. 3b). This gradual shift in molecular weight is somewhat characteristic of controlled depolymerizations, whereby termination events prevent complete unzipping of the polymer chains, as seen in optimized RAFT systems[33]. However, plotting $M_n$ and conversion of our depolymerization versus that for a theoretical controlled depolymerization shows significant deviation, indicating the loss of many monomer units per termination event (Fig. S12). The reason that lower $M_w$ chains (<15 kDa) show minimal gradual mass loss is due to the simultaneous operation of two depolymerization mechanisms: chain scission and unzipping. At low molecular weight, the rate of unzipping becomes dominant over chain scission due to the lower number of bonds per chain[34], and termination events occur significantly less often than depropagation. Therefore, the mass distribution exhibits minimal shift to lower $M_w$ as the majority of chains fully unzip.

To prove that a new phenomenon was being observed, we wanted to eliminate previously reported depolymerization chemistries. The absence of RAFT end groups in our consumer PMMA was proven through studies with RAFT-synthesized PMMA (Fig. 2C). Unlike consumer PMMA, synthesized RAFT PMMA (Figs. S15 and S16) depolymerized in the absence of UV irradiation. In addition, UV-Vis

spectroscopy showed no RAFT group absorbance (Fig. S29). The question remains as to whether other end-group functionalities, such as alkenes, could provide the handle for depolymerization[16]. Traditionally, PMMA termination during synthesis is attributed to disproportionation and recombination, with disproportionation leading to a 1:1 ratio of alkane and alkene chain ends[13]. At most, this would leave 50% of chains available to react through alkene functionality, which cannot explain the observed conversion of all chains as seen in our system.

Despite its apparent simplicity, this phenomenon had not been previously observed. Notably, a control reaction of non-RAFT PMMA under UV irradiation and an inert $N_2$ atmosphere by Anastasaki et al. reported minimal (3%) conversion at 120 °C in dioxane[33]. In contrast, at 120 °C, we observed a conversion of around 11% in dichlorobenzene. This observation led us to further investigate the effect of solvent on depolymerization.

It was hypothesized that solvents could facilitate depolymerization through one of two effects: solvents could facilitate a homolysis reaction under UV irradiation to generate chain radicals (e.g., via a solvent-mediated HAT process); or it could reduce the polymer ceiling temperature ($T_c$) through favorable interactions with monomer. Odelius et al. have recently shown that in the ring-closing depolymerization of polylactide (PLA), the $T_c$ is significantly affected by solvent selection, with PLA in DMF and DMSO exhibiting a $T_c$ almost 200 K lower than in chlorobenzene[35]. To explore these hypotheses, we screened a range of high-boiling solvents for UV-facilitated PMMA depolymerization at both 140 °C and 175 °C (Fig. 3a). Optimal results were achieved with aromatic solvents such as dichlorobenzene and diphenyl ether, while depolymerization in polar aprotic solvents such as N-methyl-2-pyrrolidone (NMP) and sulfolane exhibited minimal conversion.

To investigate whether aromatic solvent interactions caused a $T_c$ depression, analogous to that seen for PLA, the $T_c$ in various solvents was determined through polymerization reactions of MMA in DCB, sulfolane, benzonitrile, and xylenes. Determining the relationship between equilibrium monomer conversion and temperature for each of these solvents (Figs. S26 and S27, Eqs. (1) and (2)) led to a calculated $T_c$ of 104–109 °C (0.2 M) across these solvents, showing no significant relationship between conversion to monomer and $T_c$. Therefore, it is suggested that the solvent effects in this system must arise from involvement in the radical homolysis or propagation reactions, rather than through a lowering of $T_c$. To probe this further, the system was studied using electron paramagnetic resonance (EPR) spectroscopy.

Figure 3d–f shows the X-band continuous wave EPR spectra, in yellow, of DCB containing 30 mmol/L of N-tert-butyl-α-phenylnitrone (PBN) as a free-radical spin trapping agent, after 60 min of continuous irradiation at room temperature. The modeled spectrum (blue) can be attributed to spectra of three distinct PBN adducts. The first (teal in Fig. 3d) is characterized by hyperfine coupling constants (hfccs) $a_{iso}(^{14}N) = 1.43$ mT, and $a_{iso}(^{1}H^{\beta}) = 0.30$ mT and corresponds to an aryl-PBN$^{\bullet}$ radical adduct. This unusually high $a_{iso}(^{1}H)$ for an aryl-PBN$^{\bullet}$ radical adduct matches observations for o-Cl aryl radicals trapped by PBN[36]. The low-intensity contribution (magenta in Fig. 3d), characterized by hfccs $a_{iso}(^{14}N) = 1.21$ mT, $a_{iso}(^{35}Cl^{\beta}) = 0.53$ mT and $a_{iso}(^{37}Cl^{\beta}) = 0.49$ mT, together with an unresolved coupling to the $a_{iso}(^{1}H)$, corresponds to the EPR spectrum of a Cl-PBN$^{\bullet}$ radical[30]. The final contribution (lime green in Fig. 3d), characterized by $a_{iso}(^{14}N) = 1.55$ mT is similar to a PBN adduct detected also by Anastasaki et al.[30] when DCB was irradiated in the presence of PBN. However, its nature and formation remain not fully understood and explained. PBN oxidation products, i.e., α-acylaminoxyl PBN, can be eliminated, as they do not exhibit $a_{iso}(^{14}N)$ comparable to our observed value of 1.55 mT[37]. It should be noted that PBN under irradiation may form radical adducts that are not the result of genuine spin trapping. Carloni et al.[38] reported that photogenerated PBN singlet or triplet excited states can very rapidly transfer an electron to DCB,

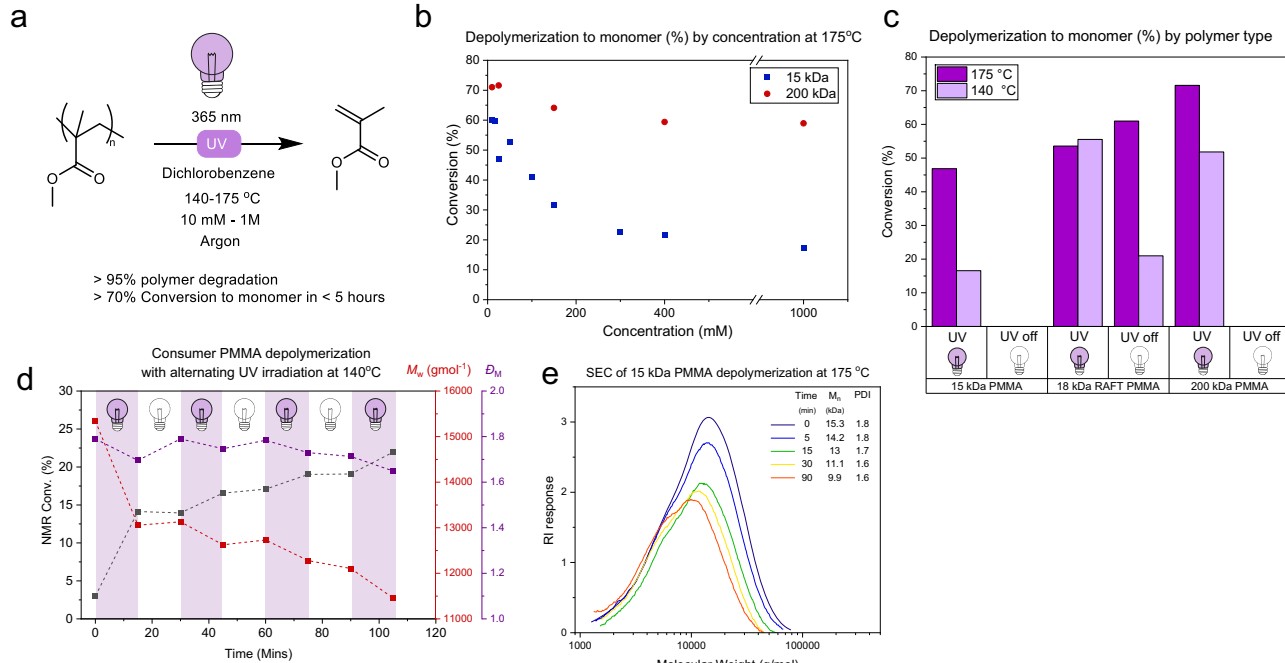

**Fig. 2 | Analysis of PMMA depolymerization under UV irradiation in dichlorobenzene. a** General scheme for the depolymerization of PMMA in dichlorobenzene. **b** $^1$H NMR conversions against repeat unit concentration for 200 kDa and 15 kDa commercial PMMA, showing increased conversion for higher molecular weight and lower concentrations (red circles−200 kDa, blue squares−15 kDa). **c** $^1$H NMR conversions for different polymers with and without UV irradiation, showing no conversion for consumer plastic without irradiation, but conversion for RAFT constructed PMMA (purple−175 °C, lilac−140 °C). **d** Polymer properties and depolymerization conversion as UV light is alternated on/off (black−monomer conversion (%), red−$M_w$, purple−polydispersity). **e** SEC (THF) chromatograms, with refractive index (RI) response, for the depolymerization of 15 kDa PMMA in DCB showing chain unzipping (navy−0 min, blue 5 min, green−15 min, yellow−30 min, red−90 min).

generating PBN$^{\cdot+}$. The latter can then react with Cl$^-$, forming the Cl-PBN$^\cdot$ spin adduct, its formation therefore being misleading on the formation of Cl$^\cdot$ and aryl$^\cdot$ radicals, if cross-checking with other spin traps is not carried out. On the basis of this, we repeated the spin trapping protocol in the presence of 5,5′-dimethyl-pyrrolidine-*N*-oxide (DMPO) as a spin trap, which does not absorb light under our irradiation conditions. The spectrum (Fig. 3e) is composed of one single spectral contribution characterized by a hfccs of $a_{iso}(^{14}N) = 1.39$ mT, and $a_{iso}(^1H^\beta) = 1.91$ mT, which matches the values reported for aryl-DMPO$^\cdot$ radical adducts in benzene. Interestingly, no Cl-DMPO$^\cdot$ radical adduct was detected, which could be explained by Cl$^\cdot$ radical annihilation being faster than the reaction of Cl$^\cdot$ with DMPO. Based on this collective evidence, we propose that an aryl-Cl bond homolysis takes place when DCB is irradiated with UV light. The resulting Cl$^\cdot$ can therefore abstract hydrogens from the PMMA backbone, activating chains for depolymerization.

Chain initiation by chlorinated aromatics provides a partial rationalization of our results, but does not explain the observation of conversion in other, non-chlorinated solvents such as benzonitrile. Therefore, an EPR spectroscopic study was also undertaken on benzonitrile, in the same conditions as PBN (Fig. 3f). The modeled spectrum (blue) consisted of two distinct spectral contributions arising from two distinct PBN-radical adducts. The first one (teal in Fig. 3f), characterized by hfccs $a_{iso}(^{14}N) = 1.40$ mT, and $a_{iso}(^1H^b) = 0.21$ mT corresponds to the EPR spectrum of an aryl-PBN$^\cdot$ radical adduct. The second contribution (lemon green in Fig. 3f), characterized by hfccs $a_{iso}(^{14}N) = 1.47$ mT, and $a_{iso}(^1H^b) = 0.194$ mT can be attributed to the spectrum of a CN- PBN$^\cdot$ radical adduct. This spectrum is characterized by a much lower signal-to-noise ratio when compared to the one in Fig. 3d, possibly indicating a much lower concentration of PBN-radical adducts being formed in the case of benzonitrile irradiation compared to DCB. Despite this, it is sufficient for CN$^\cdot$ and possibly aryl$^\cdot$ to facilitate hydrogen abstraction in a similar process to that observed in DCB.

Based on depolymerization conversions, it is postulated that many aromatic-X bonds (Br, OPh) can be cleaved in a range of substituted aromatic solvents to produce species capable of hydrogen abstraction from polymer backbones.

These EPR results also help explain differences in depolymerization behavior observed by SEC. When studying the depolymerization of 200 kDa PMMA in benzonitrile, full unzipping of polymer chains occurs, with molecular weights dropping below SEC detection limits and no significant shifts in $M_n$ or $M_w$ (Fig. 3c). This contrasts with the gradual decrease in $M_n$ observed in DCB (Fig. 3b), indicating that reaction components in the DCB system must be causing termination. With EPR studies indicating that Cl• radicals are present, it is hypothesized that these Cl• radicals can also react with polymer chain radicals, terminating chain unzipping. This depolymerization behavior is similar to that reported by Ouchi et al. with PMMA containing pendant in-chain chlorine atoms, supporting this mechanism[39]. As previously noted, EPR spectra indicated a lower amount of trapped solvent radicals in the benzonitrile solvent. This explains the high degree of unzipping behavior observed in benzonitrile (Fig. 3c) as there are fewer radicals available to cause terminaione, corroborating our suspicion of radical termination in DCB depolymerizations.

To further prove that chlorine is terminating chain depropagation, depolymerization of 200 kDa PMMA was terminated after 60 min, and precipitated. IR analysis of this polymer showed a C-Cl absorbance at 730 nm$^{-1}$ (Fig. S10), which is not observed in the PMMA beforehand, nor in DCB. For further proof, the pendant chlorine was reacted with an acridine orange dye. A small-molecule study showed the dye reacts with 2-chloroethanol under basic conditions. So, under these conditions, the PMMA was depolymerized for 5 min and then was precipitated and reacted with acridine orange at 90 °C for 18 h. The resultant polymer was precipitated and analyzed by UV-RI-SEC (Fig. 4c). The UV ($\lambda = 430$ nm) and RI traces for this species overlap

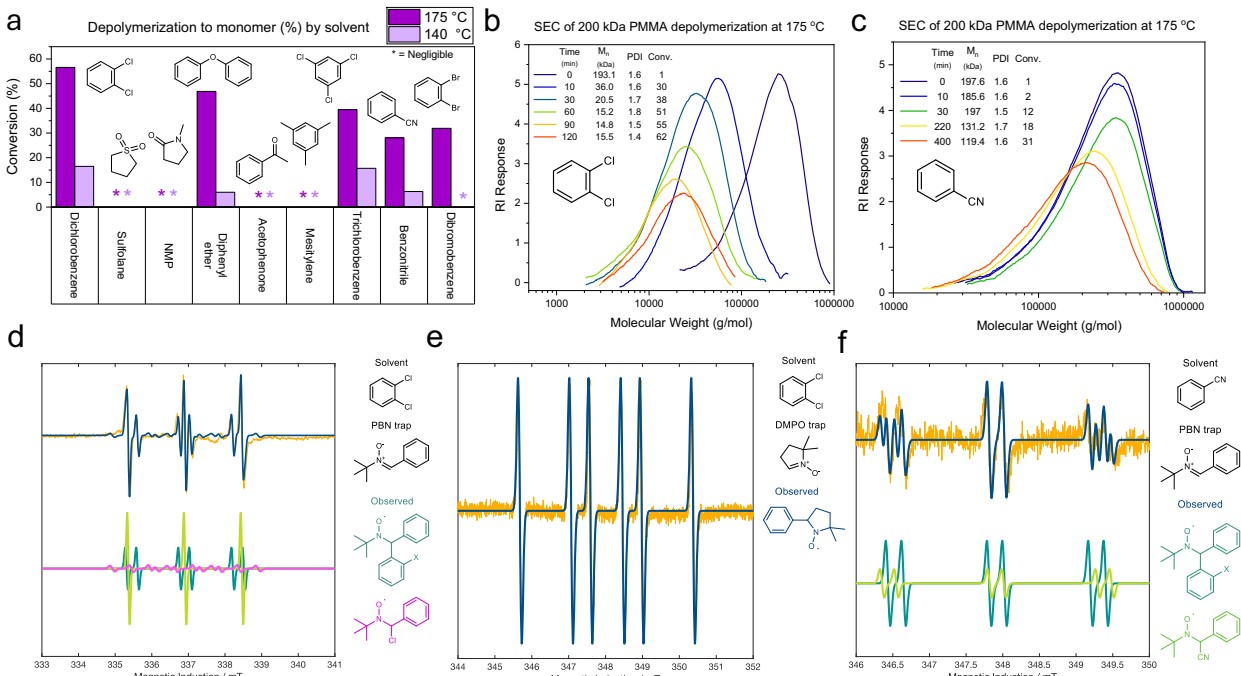

**Fig. 3 | Solvent dependency of PMMA depolymerization revealed by monomer conversions, SEC, and EPR Analyses. a** $^1$H NMR conversions of 15 kDa consumer PMMA depolymerizations in selected solvents (purple−175 °C, lilac−140 °C, * = negligible conversion, NMP *N*-methylpyrrolidone). **b** and **c** SEC (THF) chromatograms for the depolymerization of 200 kDa PMMA in **b** DCB (navy−0 min, blue 5 min, turquoise−30 min, green−60 min, yellow−90 min, red−120 min), showing gradual $M_w$ reduction and, **c** in benzonitrile (navy−0 min, blue 10 min, green−30 min, yellow−220 min, red−400 min), showing complete unzipping of polymer chains. **d** to **f** X-band continuous wave EPR spectra (yellow) of DCB (**d**, **e**) and benzonitrile (**f**) irradiated for 60 min at room temperature with a high-power (300 W) broadband (UV-A to IR) light source, in the presence of PBN (*N*-tertbutyl-phenylnitrone−**d**, **f**) and DMPO (5,5-dimethyl-1-pyrroline-*N*-oxide−**e**) spin traps. Combined modeled spectrum are overlaid (blue), and the individual contributions from species are shown in green, magenta, and teal. Spectra were recorded at room temperature, 6 mW microwave power, 100 kHz field modulation frequency, 0.1 mT field modulation amplitude, $10^5$ receiver gain, 10.24 ms time constant, 20.48 ms conversion time, and 10 spectra accumulations.

at the same retention time, whereas a control with the consumer PMMA showed negligible UV absorbance in the polymer region (Figs. S8 and S9). With Cl-PMMA in hand, it was decided to investigate chain extension through ATRP with another block of MMA monomer. However, under conditions that successfully chain-extended PMMA made through ATRP, no extension could be achieved with our depolymerized PMMA (Figs. S22 and S23). This suggests that chlorine termination does not result in an activated chlorine on the α-carbon to the carbonyl, as produced by ATRP and required for chain extension, but rather results in a less active secondary chlorine at the β-carbon.

As a result, we propose an updated mechanism for low-temperature UV-mediated PMMA depolymerization (Fig. S28) in which chain initiation can take place through HAT by solvent radicals formed by UV irradiation. Termination events occur when high concentrations of solvent radicals (i.e., Cl·) are present, preventing full unzipping by capping polymer chains with reactive handles. Small additions (10−20 vol%) of aliphatic solvent to benzonitrile lead to negligible monomer evolution, and furthermore, the addition of 10 vol % sulfolane to DCB reduces monomer evolution by over 25% (Fig. S11). These results suggest that aliphatic protons are actively inhibiting or terminating depropagation, potentially explaining the lack of conversion in solvents with abstractable aliphatic protons.

To show the applicability of this photo-initiated depolymerization to real-waste acrylic, a range of samples of Perspex® (a.k.a Plexiglass®) from waste laser cutting were subjected to our process conditions. Samples were analyzed by SEC before depolymerization, which showed that clear Plexiglass/Perspex® had an $M_n^{SEC}$ 60.8 kg mol$^{-1}$ and that blue and violet Plexiglass/Perspex® both exhibited $M_n^{SEC}$ 659 kg mol$^{-1}$ (Fig. S25). After depolymerization (Fig. 5), the violet plastic exhibited a degradation to small molecules (<1000 Da) by SEC

of over 95%, the method relied upon by Anastasaki et al.[30], which translates to a monomer conversion by $^1$H NMR of >65% (175 °C, 5 h). Clear Plexiglass/Perspex® showed slightly lower conversion, as expected due to its lower molecular weight (>50%, 175 °C, 5 h). However, the blue Perspex® showed poorer conversion to monomer despite similar molecular weight (>40%, 175 °C, 5 h), especially at 140 °C, suggesting that factors other than molecular weight can influence conversion. In this case, for example, additives such as blue dyes could be inhibiting UV penetration of the solution.

To demonstrate chemical recycling, it is important to show that the MMA produced can be repolymerized to PMMA under standard conditions at the gram scale. To achieve this, 5 g of Perspex® Frost − Violet was depolymerized for 5 h in DCB at 175 °C (reaching 55% conversion by $^1$H NMR spectroscopy). After the reaction, the solution was placed in a modified Dean-Stark apparatus, and MMA was distilled and collected (Fig. S4). Due to the close boiling points of DCB and monomer, the MMA was isolated as 2.7 mL of 73 mol% MMA in DCB (by $^1$H NMR) or 1.84 g of MMA (46% recovery from PMMA). This solution was then polymerized, uncontrolled, and initiated by AIBN at 70 °C overnight (Fig. S19). The resultant polymer was precipitated into ether and formed a glassy white solid. $^1$H NMR and SEC showed pure rPMMA, with 99% conversion, $M_w^{SEC}$ of 38.8 kDa, and polydispersity of 1.65. A controlled RAFT polymerization of the rMMA targeting 15 kDa, also successfully produced rPMMA in 99% conversion, $M_w^{SEC}$ of 13 kDa, and polydispersity of 1.11 (Figs. S17 and S18). This recycling process could be improved further, through increased temperature or by continuous distillation during the depolymerization, to recover greater amounts of pure MMA.

To test the suitability of the process for other commodity vinyl plastics, commercial polystyrene ($M_n^{SEC}$ 102 kg mol$^{-1}$) was subjected to

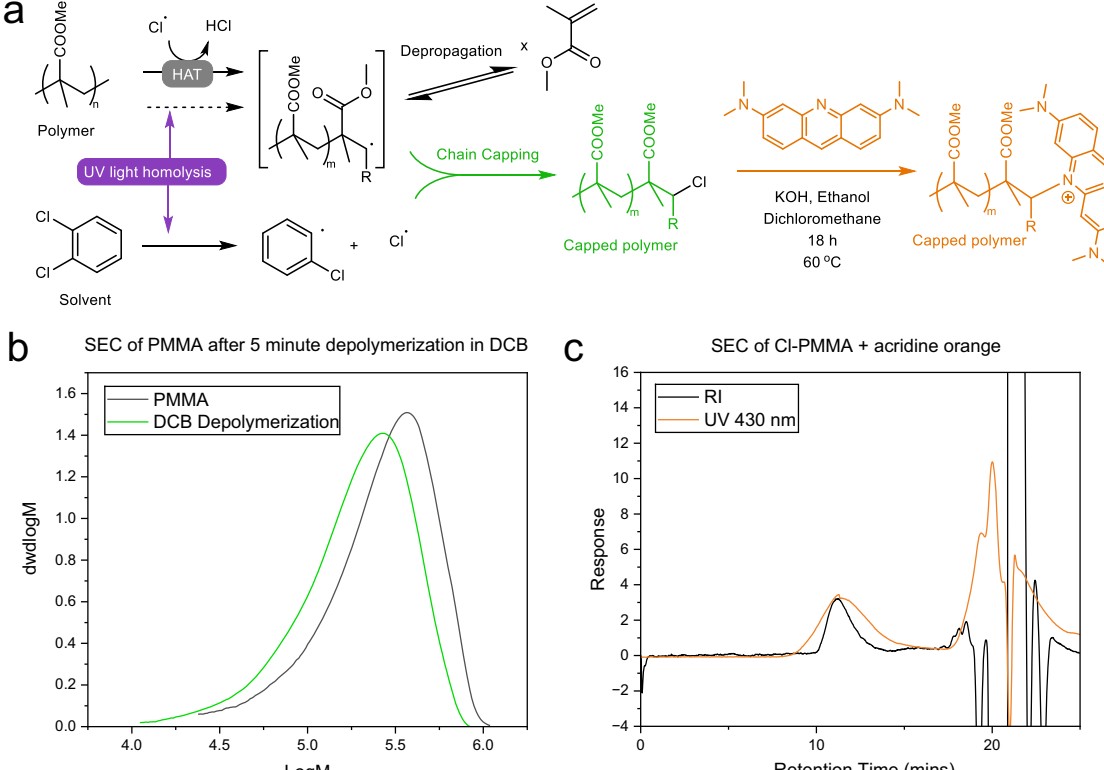

**Fig. 4 | Proposed termination pathways in the dichlorobenzene PMMA depolymerization. a** Hypothesized scheme for the initiation and termination of PMMA depolymerization by chlorine radicals and subsequent functionalization by acridine orange. The dashed line represents PMMA homolysis, which has not been ruled out. As the location of abstraction is currently unknown, R represents either the remaining polymer chain or the chain end functionality. **b** SEC (THF) chromatograms for the 5-min depolymerization of PMMA in DCB (green) against the original PMMA (black). **c** SEC (THF) chromatograms for the acridine orange tagged PMMA illustrating overlap of RI (black) and UV absorbance (430 nm, orange) of the reacted PMMA.

the same conditions in DCB at 160 °C for 4 h. SEC analysis showed an 8-fold reduction in $M_w$ and retention of dispersity (Fig. 6); however, minimal monomer formation was observed. This observation suggests that chain scission mediated by light irradiation is possible in other classes of vinyl plastics, and with optimization, monomer recovery for other consumer plastics may be achievable.

In summary, this study demonstrates the UV-mediated depolymerization of PMMA under conditions that enable controlled monomer conversion in excess of 70%. Mechanistic investigations reveal that solvent choice is key to this phenomenon, with aromatic solvents such as dichlorobenzene or benzonitrile required. It is hypothesized that solvent radicals generated by UV irradiation of photolyzable aromatics such as DCB and benzonitrile, provide HAT initiation of polymer chains leading to depolymerization above the $T_c$. In particular, demonstration of scalable depolymerization in non-chlorinated solvents paves the way for the development of more sustainable processes by avoiding the use of chlorinated solvents, which are undesirable due to their high greenhouse gas potential and environmental persistence[40,41]. High radical concentrations (i.e., in the irradiation of DCB forming Cl·) lead towards a more controlled depolymerization through termination of propagation, forming chlorine pendant PMMA. These chlorine groups are shown to react with a nucleophilic dye, proving they can be utilized for introducing functionality. Finally, successful depolymerization and subsequent repolymerization of real Plexiglass/Perspex® waste validated this process as a viable chemical recycling strategy. These findings not only provide a promising route for consumer PMMA recycling, upcycling, and derivatization but also lay the groundwork for future research aimed at understanding this approach towards the depolymerization of other plastics. Furthermore, if the termination rate can be tuned to match the depropagation rate, then pseudo-controlled depolymerization may be realized without traditional CRP techniques, potentially leading to alternative strategies for controlling polymer architectures, enabling precise tuning of molecular weights, end groups, and compositions during chemical recycling processes. Future work will target these avenues of investigation and will also focus on holistic benchmarking of this approach against pyrolysis, in terms of both economic viability and sustainability.

## Methods
### Materials and methods
All chemicals were purchased from either Sigma Aldrich or Fisher Scientific. PMMA controls were purchased from Sigma Aldrich in 15 kDa and 350 kDa molecular weights.

Samples of Perspex® were obtained from Perspex Distribution Ltd: Perspex® frost Aurora Violet S2 7T58 3 mm, Perspex® XT clear 0X00 3 mm, and Perspex® spectrum LED blue 7TL1 3 mm. Samples were broken into pieces sub 5 mm in diameter, before dissolution in solvent and depolymerization.

All NMR spectra were acquired using a 500 MHz ($^1$H) Bruker Avance III spectrometer, unless otherwise stated. CDCl$_3$ and acetone-$d_6$ were purchased from Sigma Aldrich and used as received as the NMR solvent.

Polymer molecular weight data were acquired using an Agilent 1260 Infinity system in THF eluent, fitted with 2 x PLgel 5 µm Mixed D (300 × 7.5 mm) columns and a PLgel Mixed Guard (50 × 7.5 mm) guard column operating at 35 °C. Analysis was followed with a fitted differential refractometer (dRI) operating at 35 °C. The system was calibrated by linear narrow molecular weight PMMA standards ranging from 260,900 to 885 g mol⁻¹. Data were processed using Agilent's GPC/

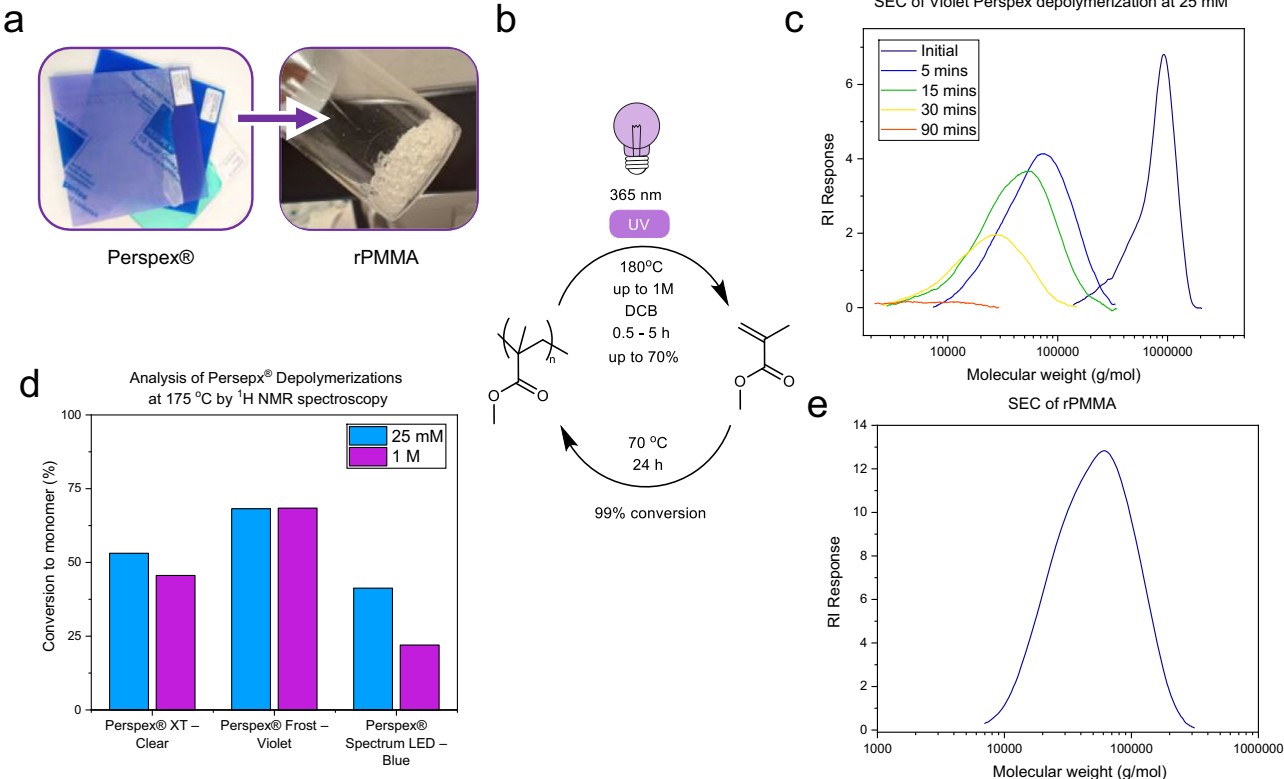

**Fig. 5 | Chemical recycling of consumer Perspex®. a** Photographs of Perspex® samples and rPMMA produced from violet Perspex®. **b** Scheme of the chemical recycling process, including repolymerization. **c** SEC (THF) chromatograms of the Violet Perspex® depolymerization (navy−0 min, blue 5 min, green−15 min, yellow− 30 min, red−90 min). **d** Depolymerization conversions measured by ¹H NMR spectroscopy for selected waste samples at 25 mM (blue) and 1 M (purple) repeat unit concentration. **e** SEC (THF) chromatogram of rPMMA.

SEC Software, Revision A.02.01. Unless otherwise stated, polymer samples were not purified prior to GPC analysis, but were filtered through a 0.5 µM filter to remove any potential particulate matter.

## PMMA depolymerization general method

PMMA was weighed into a Schlenk flask with a stirrer bar, and solvent was added. The solution was bubbled with argon flow for around 5 min and then sealed under a strong flow of argon. For high concentrations of PMMA (>250 mM), this step requires heating to around 60−80 °C to get full dissolution of the polymer. Once fully dissolved, the Schlenk flask was sealed and placed in a preheated oil bath at the temperature for the study, and a 30 W 365 nm LED lamp was used to irradiate the samples, placed ~3 cm from the flask (Fig. S1). Lamp irradiance of the sample was calculated to be approximately 100 mW/cm² (Fig. S2). After an elapsed time period, the Schenk flask was placed into a beaker of water to cool before either taking aliquots for analysis under argon or being opened to air. Details of analysis and other methods are given in the supplementary information, including the methodology for determining conversion by ¹H NMR spectroscopy using Eqs. (3) and (4) (Fig. S3).

## Original HAT experiment with xanthyl N-functionalized amide hydrogen atom transfer (HAT) catalyst

Consumer PMMA was weighed into a Schlenk flask with a stirrer bar, followed by 1−10 equivalents of xanthylation catalyst (*N*-(*tert*-Butyl)-*N*-((ethoxycarbonothioyl)thio)-3,5-bis(trifluoromethyl)benzamide) with respect to polymer chains (Fig. S5). Dichlorobenzene was then added. Dibromoanisole was occasionally added as an internal NMR standard. The solution was bubbled with argon flow for around 5 min and then sealed under a strong flow of argon. Once fully dissolved and sealed, the Schlenk flask was placed in a preheated oil bath at the temperature for the study, and a 30 W 365 nm LED array lamp was used to irradiate

the samples, placed 3-4 cm from the Schlenk flasks. After an elapsed time period, the Schlenk flask was removed and placed into a beaker of water to cool before either taking aliquots for analysis under argon, or being opened to air.

## Reaction of DCB depolymerized PMMA with acridine orange

Step 1. 400 mg of 200 kDa PMMA was dissolved in 20 mL dichlorobenzene in a Schlenk flask and degassed with argon bubbling for 10 min. The Schlenk flask was sealed and placed in a preheated oil bath at 175 °C, and a 30 W 365 nm LED lamp was used to irradiate the samples, placed ~3 cm from the flask. After 5 min, the Schlenk flask was placed into a beaker of water to cool. Once cool, the DCB solution was precipitated into a 1:1 MeOH:H₂O solution, and the precipitate was dried for use in step 2 (Fig. S7).

Step 2. 50 mg of precipitated PMMA from step 1 was dissolved in dichloromethane (5 mL). To this, 5 mg of acridine orange was added. Finally, 100 µL of a 0.5 M KOH in methanol solution was added. The solution was heated to 60 °C and left overnight (Fig. S7).

As a control, step 2 was repeated with untreated 200 kDa PMMA.

## Polymerizations

**15 kDa RAFT MMA polymerization.** Methyl methacrylate was first filtered through a plug of alumina to remove the inhibitor. Three grams of this were added to 5 mL of DCB in a Schlenk flask. To this, 51 mg of 4-Cyano-4-[(dodecylsulfanylthiocarbonyl)sulfanyl]pentanoic acid was added, followed by 100 µL of a 78 mg/mL stock solution of AIBN in DCB. The solution was bubbled with argon, and the flask was purged and sealed. The Schlenk flask was immersed in an oil bath at 70 °C and left for 48 h (Fig. S15). The resultant thick oil was sampled for conversion by ¹H NMR, determined to be 99%, and was then precipitated into diethyl ether, filtered, and dried in vacuo, to yield 2.45 g of pure

## a

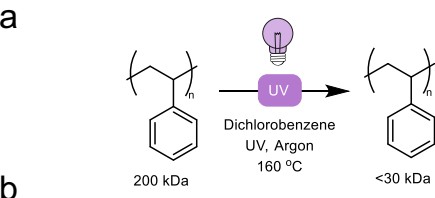

200 kDa → <30 kDa

Dichlorobenzene
UV, Argon
160 °C

## b

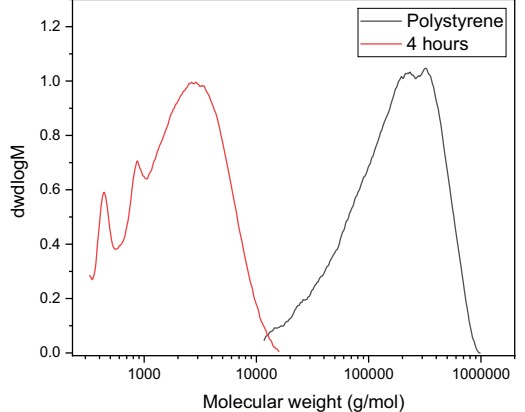

SEC of attempted polystyrene depolymerization with UV at 160 °C

**Fig. 6 | Attempted depolymerization of polystyrene. a** Attempted depolymerization of polystyrene in dichlorobenzene with UV irradiation. **b** SEC (THF) chromatogram of the virgin polymer (black) and product after reaction (red), showing a large molecular weight decrease.

PMMA (SEC: $M_n^{SEC}$ of 16.9 kg mol$^{-1}$, $Đ = 1.07$, $^1$H NMR ($d_6$-acetone, 500 MHz, Fig. S16): $M_n^{NMR}$ of 19 kg mol$^{-1}$, DP$^{NMR}$ 185).

**15 kDa RAFT rMMA polymerization.** 1 mL of recovered rMMA distillate was added to a Schlenk flask as is. To this, 17 mg of 4-cyano-4-[(dodecylsulfanylthiocarbonyl)sulfanyl]pentanoic acid was added, followed by 100 μL of a 15 mg/mL stock solution of AIBN in DCB. The solution was bubbled with argon, and the flask was purged and sealed. The Schlenk flask was immersed in an oil bath at 70 °C and left for 48 h (Fig. S17). The resultant thick oil was sampled for conversion by $^1$H NMR (Fig. S18), determined to be 96%, and was then precipitated into diethyl ether, filtered, and dried in vacuo, to yield 1720 mg of PMMA (SEC: $M_n^{SEC}$ of 20 g mol$^{-1}$, $Đ = 1.11$, $M_n^{NMR}$ of 13 kg mol$^{-1}$, DP$^{NMR}$ 125).

**Uncontrolled FRP of rMMA.** 0.5 mL of recovered rMMA distillate was added to a Schlenk flask as is. AIBN was added to the solution by adding 100 μL of a 10 mg/mL stock solution in DCB. The solution was bubbled with argon, and the flask was purged and sealed. The Schlenk flask was immersed in an oil bath at 70 °C and left for 24 h (Fig. S19). The resultant thick oil was sampled for conversion by $^1$H NMR, determined to be 96%, and was then precipitated into diethyl ether, filtered, and dried in vacuo, to yield 458 mg of pure PMMA (SEC: $M_n^{SEC}$ of 38.8 kg mol$^{-1}$, $Đ = 1.65$).

**ATRP MMA polymerization.** Methyl methacrylate was first filtered through a plug of alumina to remove the inhibitor. 2 mL (1.88 g, 18.7 mmol) of this monomer was added to a 30 mL glass vial along with tosyl chloride (0.018 g, 0.094 mmol), AIBN (0.002 g, 0.009 mmol), 30 μL of a 1 mL stock solution of tris(2-pyridylmethyl)amine (TPMA, 0.032 g, 0.112 mmol) and copper (II) chloride (0.005 g, 0.037 mmol) in DMF and 2 mL of anisole. The solution was purged with nitrogen for 10 min and sealed before being immersed in an oil bath at 65 °C for 24 h (Fig. S20). The resulting solution was sampled for monomer conversion by $^1$H NMR (CDCl$_3$) and determined to be 85%. The polymer

was then precipitated into methanol, filtered, and dried in vacuo to yield 0.46 g of PMMA-Cl (SEC: $M_n^{SEC}$ of 58.0 kg mol$^{-1}$, $Đ = 1.43$).

**Chain extension of ATRP PMMA.** Methyl methacrylate was first filtered through a plug of alumina to remove the inhibitor. 0.08 mL (0.075 g, 0.75 mmol) of this monomer was added to a 30 mL glass vial along with the PMMA-ATRP macroinitiator (0.075 g), AIBN (0.002 g, 0.009 mmol), 30 μL of a 1 mL stock solution of tris(2-pyridylmethyl) amine (TPMA, 0.032 g, 0.112 mmol) and copper (II) chloride (0.005 g, 0.037 mmol) in DMF and 2 mL of anisole. The solution was purged with nitrogen for 10 min and sealed before being immersed in an oil bath at 65 °C for 24 h (Fig. S21). The resulting solution was sampled for monomer conversion by $^1$H NMR (CDCl$_3$) and determined to be 85%. The polymer was then precipitated into methanol, filtered, and dried in vacuo to yield 0.097 g of polymer (SEC: $M_n^{SEC}$ of 148.2 kg mol$^{-1}$, $Đ = 1.57$).

**Attempted chain extension of Cl-PMMA.** PMMA-Cl from Step 1 of the 10-min DCB PMMA depolymerization was subjected to the same chain extension procedure as for ATRP PMMA above (Fig. S22). However negligible change in $M_n$ was observed by SEC for this (Fig. S23), suggesting the chlorine is not at the alpha-position relative to the carbonyl as seen in the ATRP PMMA.

**Attempt to depolymerize polystyrene.** Polystyrene ($M_n^{SEC}$ of 102 kg mol$^{-1}$, $Đ = 2.19$) was weighed into a Schlenk flask with a stirrer bar, and solvent was then added. The solution was bubbled with argon flow for around 5 min and then sealed under a strong flow of argon (Fig. S13). The Schlenk flask was then placed in a preheated oil bath at 160 °C, and a 30 W 365 nm LED array lamp was used to irradiate the sample. After 4 h, the Schlenk flask was removed and placed into a beaker of water to cool before being opened to air. The solution was analyzed by THF-SEC and $^1$H NMR spectroscopy, and despite a large $M_w$ decrease observed by SEC (Fig. S14), minimal production of monomer could be observed by $^1$H NMR spectroscopy. It is postulated that through optimization, unzipping to monomer can be achieved.

## Data availability

All experimental and spectroscopic results are available in the Supplementary Information, including tabulation of all reactions performed. All data are available from the corresponding authors upon request.

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

## Acknowledgements

This work was financially supported by the Research England Development Fund and EPSRC grant number EP/Z532782/1 [Innovation Centre for Applied Sustainable Technologies (iCAST) to MGD, SF, JH and GI and Sustainable Chemicals and Materials Manufacturing Hub (SCHEMA) to MGD, respectively]. The team and facilities at the Innovation Center for Sustainable Technologies (iCAST) and the Institute of Sustainability and Climate Change (ISCC) at the University of Bath are thanked for hosting and supporting the research program. Dr Alexander J. Cresswell, Dr Tim Woodman and Dr Martin Levere of the University of Bath and the research facilities at the University of Bath (https://doi.org/10.15125/mx6j-3r54) are thanked for their technical provision, support, and assistance throughout this work.

## Author contributions

Conceptualization: J.T.H. Methodology: J.T.H., A.F., M.G.D., S.F. Investigation: J.T.H., C.R.M., A.F., G.I. Visualization: J.T.H., A.F. Funding acquisition: M.G.D. Project administration: M.G.D., S.F. Supervision: S.F., A.F., M.G.D. Writing—original draft: J.T.H., A.F. Writing—review and editing: all authors.

## Competing interests

The authors declare no competing interests.
