## [Transparent Peer Review file · Nature Communications]

Photo-initiated solvent-mediated depolymerization of consumer poly(methyl methacrylate) without chlorinated reagents

Corresponding Author: Dr Jonathan Husband

Version 0:

Reviewer comments:

Reviewer #1

(Remarks to the Author)

The manuscript by Husband et. al. reports a recycling method for PMMA plastics using photochemistry with heating. The importance of the topic cannot be understate given the buildup of plastic waste. The method is advantageous compared to conventional pyrolysis under 350-400 C. Using 365 nm UV light at 120-180 degree C, the researchers observed radicals (via EPR) from aromatic solvents and utilized them for H atom abstraction from PMMA to initiate depolymerization. Given the recently published work that found chlorine radical was essential, this work shows that other radical sources can accomplish similar reactivity. The study is scientifically and experimentally thorough.

However, as presented, this work isn't sufficiently novel for publication in Nature Communications. This specific approach was demonstrated in the previously published paper (Science 2025). Although the researchers were able to obtain 90% monomer yield, the use of UV light and large volume of organic solvent raise safety concern in industry. The key argument regarding "Cl not required" is valid – industries intentionally avoid the use of halogenated solvents, but it's not clear that the other aromatic solvents (with the combination of UV light) provide a reasonable alternative to existing industrial technology. Additionally, most of the manuscript still focuses on using dichlorobenzene and demonstrates that it is superior to other solvents. The general approach of using exogenous radicals or cleavage chain ends (or monomers) has been well-demonstrated in the following works (Science 2025, 387, 874; JACS 2022, 144, 4678; ACIE 2023, e202313232; Chem 2024, 10, 388, JACS 2024, 146, 18848; and more). While I think this paper is well-constructed and informative, it currently doesn't meet the criteria of novelty for publishing in Nature Communications. The authors do initially mention an alternative approach of using an exogenous HAT agent with dichlorobenzene – does this work with the more industry compatible solvents with comparable or better efficiency to DCB? Having a cooperative background radical generation from solvent (not DCB) wouldn't be deleterious to the impact.

Reviewer #2

(Remarks to the Author)

The authors have addressed all comments from the reviewers. It is ready for publishing.
